# UniFL: Improve Latent Diffusion Model via Unified Feedback Learning

Jiacheng Zhang[1,2,‡]    Jie Wu[2,†,‡,*]    Yuxi Ren[2]    Xin Xia[2]    Huafeng Kuang[2]
Pan Xie[2]    Jiashi Li[2]    Xuefeng Xiao[2]
Weilin Huang[2]    Shilei Wen[2]    Lean Fu[2]    Guanbin Li[1,3*]

[1]Sun Yat-sen University    [2]Bytedance Inc    [3]Peng Cheng Laboratory
Project Page: https://uni-fl.github.io/

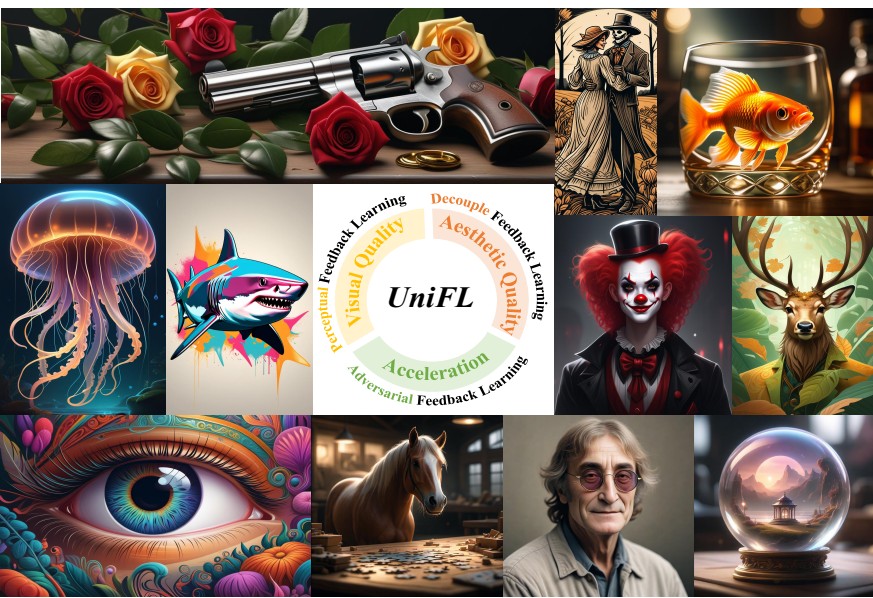

Figure 1: Generated samples with 20 steps inference from `stable-diffusion-xl-base-1.0` optimized by Unified Feedback Learning (UniFL). The last three images of the third row are generated with 4 steps.

## Abstract

Latent diffusion models (LDM) have revolutionized text-to-image generation, leading to the proliferation of various advanced models and diverse downstream applications. However, despite these significant advancements, current diffusion models still suffer from several limitations, including inferior visual quality, inadequate aesthetic appeal, and inefficient inference, without a comprehensive solution in sight. To address these challenges, we present **UniFL**, a unified framework that leverages feedback learning to enhance diffusion models comprehensively. UniFL stands out as a universal, effective, and generalizable solution applicable to various diffusion models, such as SD1.5 and SDXL. Notably, UniFL consists of three key components: perceptual feedback learning, which enhances visual quality; decoupled feedback learning, which improves aesthetic appeal; and adversarial feedback learning, which accelerates inference. In-depth experiments and extensive user studies validate the superior performance of our method in enhancing generation quality and inference acceleration. For instance, UniFL surpasses ImageReward by 17% user preference in terms of generation quality and outperforms LCM and SDXL Turbo by 57% and 20% general preference with 4-step inference.

*Corresponding authors: wujie10558@gmail.com ,liguanbin@mail.sysu.edu.cn ; †: project lead ; ‡: Equal Contribution. Work done during an internship at ByteDance.

38th Conference on Neural Information Processing Systems (NeurIPS 2024).

# 1 Introduction

The emergence of diffusion models has led to remarkable advances in the field of text-to-image (T2I) generation, marked by notable milestones like DALLE-3 [1], Imagen [2], Midjourney [3], etc, elevating the generation quality of images to an unprecedented level. Particularly, the introduction of open-source image generation models, exemplified by latent diffusion model (LDM) [4], has inaugurated a transformative era of text-to-image generation, triggering numerous downstream applications such as T2I personalization [5, 6, 7, 8], controllable generation [9, 10, 11] and text-to-video (T2V) generation [12, 13, 14]. Nevertheless, despite these advancements achieved thus far, current latent diffusion-based image generation models still exhibit certain limitations. i) Inferior visual quality: The generated images still suffer from poor visual quality and lack authenticity. Examples include characters with incomplete limbs or distorted body parts, as well as limited fidelity in terms of style representation. ii) Inadequate aesthetic appeal: The generated image tends to lack aesthetic appeal and often fails to align with human preferences, especially in the abstract aesthetic concepts aspects such as color, lighting, atmosphere, etc. iii) Slow inference speed: The iterative denoising process employed by diffusion models led to inefficiencies during inference that significantly impede generation speed, thereby limiting the practicality of these models in various application scenarios. Recently, numerous works have endeavored to address the aforementioned challenges. For instance, RAPHAEL [15] resorts to the techniques of Mixture of Experts) [16, 17, 18] boost the generation performance via stacking the space MoE and time MoE block. Works [19, 20, 21, 22, 23] represented by ImageReward [23] propose incorporating human preference feedback to guide diffusion models toward aligning with human preferences. SDXL Turbo [24], PGD [25], and LCM [26, 27], on the other hand, targets on achieve inference acceleration through techniques like distillation and consistency models [28]. However, these methods primarily concentrate on tackling individual problems through specialized designs, which poses a significant challenge to the elegant integration of these techniques. For example, MoE significantly complicates the pipeline, making the acceleration method infeasible to apply, and the consistency models [28] alter the denoising process of the diffusion model, making it arduous to directly apply the ReFL preference tuning framework proposed by ImageReward [23]. Therefore, a natural question arises: *Can we devise a more effective approach that comprehensively enhances diffusion models in terms of image quality, aesthetic appearance, and generation speed?*

To tackle this issue, we present UniFL, a solution that offers a comprehensive improvement to latent diffusion models through unified feedback learning formulation. UniFL aims to boost the visual generation quality, enhance aesthetic attractiveness, and accelerate the inference process. To achieve these objectives, UniFL features three novel designs upon the unified formulation of feedback learning. Firstly, we introduce a pioneering perceptual feedback learning (PeFL) framework that effectively harnesses the extensive knowledge embedded within diverse existing perceptual models to provide more precise and targeted feedback on the potential visual defects of the generated results. Secondly, we employ decoupled aesthetic feedback learning to boost the visual appeal, which breaks down the coarse aesthetic concept into distinct aspects such as color, atmosphere, and texture, simplifying the challenge of abstract aesthetic optimization. Furthermore, an active prompt selection strategy is also introduced to choose the more informative and diverse prompt to facilitate more efficient aesthetics preference learning. Lastly, UniFL develops adversarial feedback learning to achieve inference acceleration by incorporating the adversarial objective in feedback tuning. We instantiate UniFL with a two-stage training pipeline and validate its effectiveness with SD1.5 and SDXL, yielding impressive improvements in generation quality and acceleration. Our contributions are summarized as follows:

- **New Insight**: Our proposed method, UniFL, introduces a unified framework of feedback learning to optimize the visual quality, aesthetics, and inference speed of diffusion models. To the best of our knowledge, UniFL offers the first attempt to address both generation quality and speed simultaneously, offering a fresh perspective in the field.

- **Novelty and Pioneering**: In our work, we shed light on the untapped potential of leveraging existing perceptual models in feedback learning for diffusion models. We highlight the significance of decoupled reward models and elucidate the underlying acceleration mechanism through adversarial training.

- **High Effectiveness**: Through extensive experiments, we demonstrate the substantial improvements achieved by UniFL across various types of diffusion models, including SD1.5 and SDXL, in terms of generation quality and inference acceleration.

## 2 Related Works

**Text-to-Image Diffusion Models.** Text-to-image generation has gained unprecedented attention over other traditional tasks [29, 30, 31, 32, 33]. Recently, diffusion models have gained substantial attention and emerged as the *de facto* mainstream method for text-to-image generation, surpassing traditional image generative models like GAN [34] and VAE [35]. Numerous related works have been proposed, including GLIDE [36], DALL-E2 [1], Imagen [2], CogView [37] etc.. Among these, Latent Diffusion Models (LDM) [4] extend the diffusion process to the latent space and significantly improve the training and inference efficiency of the diffusion models, opening the door to diverse applications such as controllable generation [9, 10], image editing [11, 38, 39], and image personalization [5, 7, 6] and so on. Even though, current text-to-image diffusion models still have limitations in inferior visual generation quality, deviations from human aesthetic preferences, and inefficient inference. The target of this work is to offer a comprehensive solution to address these issues.

**Improvements on Text-to-Image Diffusion Models.** Given the aforementioned limitations, researchers have proposed various methods to tackle these issues. Notably, [40, 15, 41] focuses on improving generation quality through more advanced training strategies. Inspired by the success of reinforcement learning with human feedback (RLHF) [42, 43] in the field of LLM, [20, 21, 44, 23, 45] explore the incorporation of human feedback to improve image aesthetic quality. On the other hand, [25, 24, 28, 27, 26] concentrate on acceleration techniques, such as distillation and consistency models [28] to achieve inference acceleration. While these methods have demonstrated their effectiveness in addressing specific challenges, their independent nature makes it challenging to combine them for comprehensive improvements. In contrast, our study unifies the objective of enhancing visual quality, aligning with human aesthetic preferences, and acceleration through the feedback learning framework.

## 3 Preliminaries

**Latent Diffusion Model.** Text-to-image latent diffusion models leverage diffusion modeling to generate high-quality images based on textual prompts, which generate images from Gaussian noise through a gradual denoising process. During pre-training, a sampled image $x$ is first processed by a pre-trained VAE encoder to derive its latent representation $z$. Subsequently, random noise is injected into the latent representation through a forward diffusion process, following a predefined schedule $\{\beta_t\}^T$. This process can be formulated as $z_t = \sqrt{\overline{\alpha}_t} z + \sqrt{1 - \overline{\alpha}_t}\epsilon$, where $\epsilon \in \mathcal{N}(0, 1)$ is the random noise with identical dimension to $z$, $\overline{\alpha}_t = \prod_{s=1}^{t} \alpha_s$ and $\alpha_t = 1 - \beta_t$. To achieve the denoising process, a UNet $\epsilon_\theta$ is trained to predict the added noise in the forward diffusion process, conditioned on the noised latent and the text prompt $c$. Formally, the optimization objective of the UNet is:

$$\mathcal{L}(\theta) = \mathbb{E}_{z,\epsilon,c,t}[||\epsilon - \epsilon_\theta(\sqrt{\overline{\alpha}_t} z + \sqrt{1 - \overline{\alpha}_t}\epsilon, c, t)||_2^2] \tag{1}$$

**Reward Feedback Learning.** Reward feedback learning (ReFL) [23] is a preference fine-tuning framework that aims to improve the diffusion model via human preference feedback. It consists of two phases: (1) Reward Model Training and (2) Preference Fine-tuning. In the Reward Model Training phase, human preference data is collected to train a human preference reward model, which serves as a proxy to provide human preferences. More specifically, considering two candidate generations, denoted as $x_w$ (preferred generation) and $x_l$ (unpreferred one), the loss function for training the human preference reward model $r_\theta$ can be formulated as follows:

$$\mathcal{L}_{\mathrm{rm}}(\theta) = -\mathbb{E}_{(c,x_w,x_l)\sim\mathcal{D}}[log(\sigma(r_\theta(c, x_w) - r_\theta(c, x_l)))] \tag{2}$$

where $\mathcal{D}$ denotes the collected feedback data, $\sigma(\cdot)$ represents the sigmoid function, and $c$ corresponds to the text prompt. The reward model $r\theta$ is optimized to produce a reward score that aligns with human preferences. In the Preference Fine-tuning phase, ReFL begins with an input prompt $c$, initializing a random latent variable $x_T$. The latent variable is then progressively denoised until reaching a randomly selected timestep $t$. Then, the denoised image $x_0'$ is directly predicted from $x_t$. The reward model obtained from the previous phase is applied to this denoised image, generating the expected preference score $r_\theta(c, x_0')$. ReFL maximizes such preference scores to make the diffusion model generate images that align more closely with human preferences:

$$\mathcal{L}_{\mathrm{refl}}(\theta) = \mathbb{E}_{c\sim p(c)}\mathbb{E}_{x_0'\sim p(x_0'|c)}[-r(x_0', c)] \tag{3}$$

Our method follows a similar learning framework to ReFL but devises several novel components to enable comprehensive improvements.

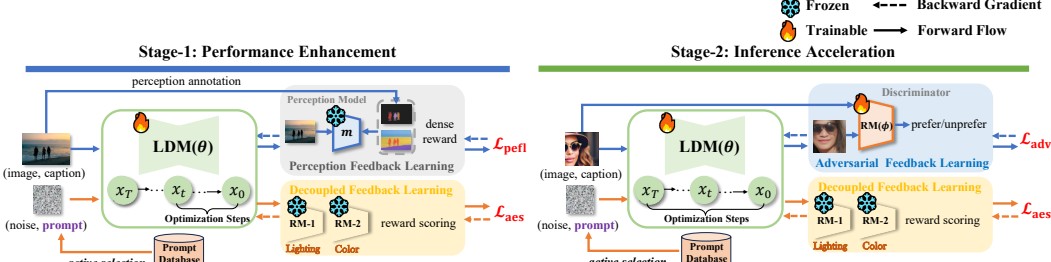

Figure 2: **Overview of *UniFL*.** We leverage a unified feedback learning framework to enhance the model performance and inference speed comprehensively. The training process of UniFL is divided into two stages, the first stage aims to improve visual quality and aesthetics, and the second stage speeds up model inference.

# 4 UniFL: Unified Feedback Learning

Our proposed method, UniFL, aims to improve the latent diffusion models in various aspects, including visual generation quality, human aesthetic quality, and inference efficiency. our method takes a unified feedback learning perspective, offering a comprehensive and streamlined solution. An overview of UniFL is illustrated in Fig.2. In the following subsections, we delve into the details of three key components: perceptual feedback learning to enhance visual generation quality (section 4.1); decoupled feedback learning to improve aesthetic appeal (section 4.2); and adversarial feedback learning to facilitate inference acceleration (section 4.3).

## 4.1 Perceptual Feedback Learning

Current diffusion models exhibit limitations in achieving high-fidelity visual generation, for example, object structure distortion. These limitations stem from the reliance on reconstruction loss(MSE loss) solely in the latent space, which lacks structural supervision on the high-level visual quality. To address this issue, we propose perceptual feedback learning (PeFL). Our key insight is that various visual perception models already embed rich visual priors, which can be exploited to provide feedback for visual generation and fine-tune the diffusion model. The complete PeFL process is summarized in Algorithm 1. In contrast to ReFL, which starts from a randomly initialized latent representation and only considers the text prompt as a condition, PeFL incorporates image content as an additional visual condition for perceptual guidance. Specifically, given a text-image pair, $(c, x)$, we first select a forward step $T_a$ and inject noise into the ground truth image to obtain a conditional latent $x_0 \rightarrow x_{T_a}$. Subsequently, we randomly select a denoising time step $t$ and denoising from $x_{T_a}$, yielding $x_{T_a} \rightarrow x_{T_a-1}... \rightarrow x_t$. Next, we directly predict $x_t \rightarrow x_0'$. By incorporating the visual condition input, the denoised image is expected to restore the same high-level visual characteristics, such as object structure, and style, which existing perception models can capture. For instance, in the case of object structure, the instance segmentation model can serve as a valuable resource as it provides essential descriptions of object structure through instance masks. Consequently, the feedback on the generation of such visual characteristics on $x_0'$ can be obtained by comparing it with the ground truth segmentation mask via:

$$\mathcal{L}_{\text{pefl}}^{\text{struct}}(\theta) = \mathbb{E}_{x_0 \sim \mathcal{D}, x_0' \sim G(x_{t_a})} \mathcal{L}_{\text{instance}}(m_I(x_0^{'}), \text{GT}(x_0)) \quad (4)$$

where $m_I$ is the instance segmentation model, $\text{GT}(x_0)$ is the ground truth instance segmentation mask and $\mathcal{L}_{\text{instance}}$ is the instance segmentation loss. Note that our PeFL differs from ReFL as indicated by the red font in Algorithm 1. With the visual condition input and perception model, the diffusion model is allowed to get a detailed and focused feedback signal on a specific aspect, instead of the general quality feedback offered by ReFL. Moreover, the flexibility of PeFL allows us to leverage various existing visual perceptual models, more examples can be found in the Appendix A.

## 4.2 Decoupled Feedback Learning

**Decoupled Aesthetic Fine-tuning.** Existing text-to-image diffusion models exhibit shortcomings in images that satisfy human aesthetic preferences. While PeFL prioritizes objective visual quality,

aesthetic quality is inherently subjective and abstract, requiring human aesthetic feedback to steer the generation process. Despite ImageReward's attempt to incorporate human aesthetic preferences through a reward model, its performance is hindered by oversimplified modeling that fails to capture the multidimensional nature of human aesthetic preferences. Generally, humans consider the aesthetic attractiveness of an image from various aspects, such as color, lighting, etc, and conflating these aspects without distinguishing during preference tuning would encounter optimization conflicts as evidenced in [46]. To address this issue, we follow [23] to achieve aesthetic preference tuning but suggest decoupling the various aesthetic aspects when constructing preference reward models. Specifically, we decomposed the general aesthetic concept into representative dimensions and collected the corresponding annotated data, respectively. These dimensions include color, layout, lighting, and detail. Subsequently, we train a separate aesthetic preference reward model for each annotated data according to Eq.2. Finally, we leveraged these reward models for aesthetic preference tuning:

$$\mathcal{L}_{\text{aes}}(\theta) = \sum_{d}^{K} \mathbb{E}_{c\sim p(c)}\mathbb{E}_{x_0'\sim p(x_0'|c)}[\texttt{ReLU}(\alpha_d - r_d(x_0', c))] \tag{5}$$

$r_d$ is the aesthetic reward model on $d$ dimension, $d \in \{\text{color}, \text{layout}, \text{detail}, \text{lighting}\}$, $\alpha_d$ is the dimension-aware hinge coefficient, and $K$ is the number of fine-grained aesthetic dimension.

**Active Prompt Selection.** We observed that when using randomly selected prompts for aesthetic preference fine-tuning, the diffusion model tends to rapidly overfit the reward model due to the limited semantic richness, leading to diminished effectiveness of the reward model. To address this issue, we further propose an active prompt selection strategy, which selects the most informative and diverse prompt from a prompt database. This selection process involves two key components: a semantic-based prompt filter and nearest neighbor prompt compression. By leveraging these techniques, the overfitting can be greatly mitigated, achieving more efficient aesthetic reward fine-tuning. More details of this strategy are presented in the Appendix.B.2.

---

**Algorithm 1** Perceptual Feedback Learning (PeFL)

---
1: **Dataset:** Captioned perceptual text-image dataset with $\mathcal{D} = \{(\text{txt}_1, \text{img}_1), ...(\text{txt}_n, \text{img}_n)\}$
2: **Input:** LDM with pre-trained parameters $w_0$, perceptual model $m_\cdot$, perceptual loss function $\Phi$, loss weight $\lambda$
3: **Initialization:** The number of noise scheduler time steps $T$, add noise timestep $T_a$, denoising time step $t$.
4: **for** perceptual data point $(\text{txt}_i, \text{img}_i) \in \mathcal{D}$ **do**
5:    $x_0 \leftarrow \text{VaeEnc}(\text{img}_i)$ // From image to latent
6:    $x_{T_a} \leftarrow \textbf{AddNoise}(x_0)$ // Add noise to latent
7:    **for** $j = T_a, ..., t + 1$ **do**
8:       **no grad:** $x_{j-1} \leftarrow \text{LDM}_{w_i}\{x_j\}$
9:    **end for**
10:    **with grad:** $x_{t-1} \leftarrow \text{LDM}_{w_i}\{x_t\}$
11:    $x_0' \leftarrow x_{t-1}$ // Predict the denoised latent
12:    $\text{img}_i' \leftarrow \text{VaeDec}(x_0')$ // From latent to image
13:    $\mathcal{L}_{\textbf{pefl}} \leftarrow \lambda\Phi(m(\textbf{img}_i'), \text{GT}(\textbf{img}_i))$ // PeFL loss by perceptual model
14:    $w_{i+1} \leftarrow w_i$ // Update $\text{LDM}_{w_i}$ using PeFL loss
15: **end for**

---

## 4.3 Adversarial Feedback Learning

The inherent iterative denoising process of diffusion models significantly hinders their inference speed. To address this limitation, we introduce adversarial feedback learning to reduce the denoising steps during inference. Specifically, to achieve inference acceleration, we exploit a general reward model $r_a(\cdot)$ to improve the generation quality of fewer denoising steps. However, as studied in [23], the samples under low inference steps tend to be too noisy to obtain the correct rewarding scores. To tackle this problem, rather than freeze the reward model during fine-tuning, we incorporate an extra adversarial optimization objective by treating $r_a(\cdot)$ as a **discriminator** and update it together with the diffusion model. Concretely, we follow a similar way with PeFL to take an image as input and execute the diffusion and denoising consecutively. Afterward, in addition to maximizing the reward score of the denoised image, we also update the reward model in an adversarial manner. The optimization objective is formulated as:

$$\begin{aligned} \mathcal{L}^G(\theta) &= \mathbb{E}_{c\sim p(c)}\mathbb{E}_{x_0'\sim p(x_0'|c)}[-r_a(x_0', c)], \\ \mathcal{L}^D(\phi) &= -\mathbb{E}_{(x_0, x_0', c)\sim\mathcal{D}_{\text{train}}, t\sim[1,T]}[\log\sigma(r_a(x_0)) + \log(1 - \sigma(r_a(x_0')))]. \end{aligned} \tag{6}$$

where $\theta$ and $\phi$ are the parameters of the diffusion model and discriminator. With the adversarial objective, the reward model is always aligned with the distribution of the denoised images with various denoised steps, enabling the reward model to function well across all the timesteps. Note that our method is distinct from the existing adversarial diffusion methods like SDXL-Turbo [24]. These

methods take the *adversarial distillation* manner to accelerate the inference, which tends to require another LDM as the teacher model to realize distillation, incurring considerable memory costs. By contrast, we follow the *reward feedback learning* formulation, which integrates adversarial training with the reward tuning and achieves the adversarial reward feedback tuning via the lightweight reward model.

### 4.4 Training Objective

We employ a two-stage training pipeline to implement UniFL. The first stage focuses on improving generation quality, leveraging perceptual feedback learning and decoupled feedback learning to boost visual fidelity and aesthetic appeal. In the second stage, we apply adversarial feedback learning to accelerate the diffusion inference speed. To prevent potential degradation, we also include decoupled feedback learning to maintain aesthetics. The training objectives of each stage are summarized as follows:

$$\mathcal{L}^1(\theta) = \mathcal{L}_{\text{pefl}}(\theta) + \mathcal{L}_{\text{aes}}(\theta); \quad \mathcal{L}^2(\theta, \phi) = \mathcal{L}^G(\theta) + \mathcal{L}^D(\phi) + \mathcal{L}_{\text{aes}}(\theta) \tag{7}$$

## 5 Experiments

### 5.1 Implementation Details and Metrics

**Dataset.** We utilized the COCO2017 [47] train split dataset with instance annotations and captions for structure optimization with PeFL. Additionally, we collected the human preference dataset for the decoupled aesthetic feedback learning from diverse aspects (such as color, layout, detail, and lighting). 100,000 prompts are selected for aesthetic optimization from DiffusionDB [48] via active prompt selection. During the adversarial feedback learning, we use data from the aesthetic subset of LAION [49] with image aesthetic scores above 5.

**Training Setting.** We utilize the SOLO [50] as the instance segmentation model. We utilize the DDIM [51] scheduler with a total of 20 inference steps. $T_a = 10$ and the optimization steps $t \in [0, 5]$ during PeFL training. For adversarial feedback learning, we initialize the adversarial reward model with the weight of the aesthetic preference reward model of details. During adversarial training, the optimization step is set to $t \in [0, 20]$ encompassing the entire diffusion process. Our training per stage costs around 200 A100 GPU hours.

**Baseline Models.** We choose two representative text-to-image diffusion models with distinct generation capacities to comprehensively evaluate the effectiveness of UniFL, including (i) SD1.5 [4]; (ii) SDXL [40]. Based on these models, we pick up several state-of-the-art methods(i.e. ImageReward [23], Dreamshaper [52], and DPO [22] for generation quality enhancement, LCM [27], SDXL-Turbo [24], and SDXL-Lightning [53] for inference acceleration) to compare the effectiveness of quality improvement and acceleration. All results of these methods are reimplemented with the official code provided by the authors.

**Evaluation Metrics.** We generate the 5K image with the prompt from the COCO2017 validation split to report the Fréchet Inception Distance (FID) [54] as the overall visual quality metric. We also report the CLIP score with ViT-B-32 [55] and the aesthetic score with LAION aesthetic predictor to evaluate the text-to-image alignment and aesthetic quality of the generated images, respectively. Given the subjective nature of quality evaluations, we further conducted comprehensive user studies to obtain a more accurate evaluation.

### 5.2 Main Results

**Quantitative Comparison.** Tab.1 summarize the quantitative comparisons with competitive approaches on SD1.5 and SDXL. Generally, UniFL exhibits consistent performance improvement on both architectures and surpasses the existing methods of focus on improving generation quality or acceleration. Specifically, for the generation quality, UniFL surpasses both DreamShaper (DS) and ImageReward (IR) across all metrics, where the former relies on high-quality training images while the latter exploits the human preference for fine-tuning. It is also the case when compared with the recently proposed preference tuning method DPO. In terms of acceleration, UniFL also exhibits notable performance advantages, surpassing the LCM with the same 4-step inference on both SD1.5 and SDXL. Surprisingly, we found that UniFL sometimes obtained even better aesthetic quality

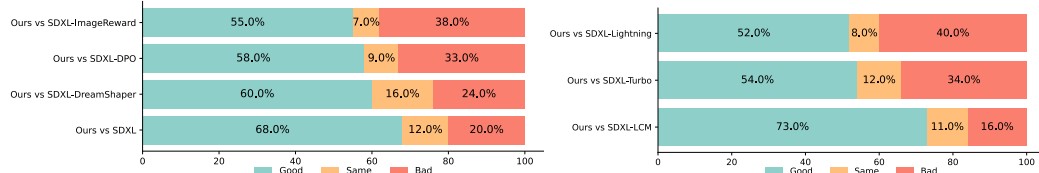

Figure 3: **User study** about UniFL and other methods with 10 users on the generation of 500 prompts in generation quality (left) and inference acceleration (right).

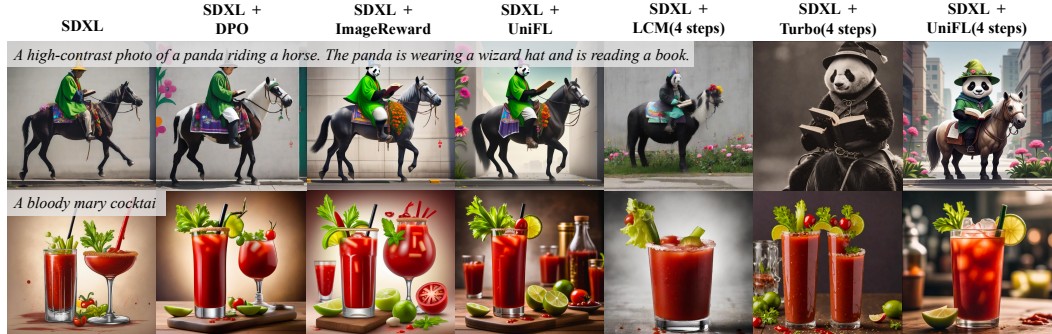

Figure 4: **Qualitive comparison** of the generation results of different methods based on SDXL.

with fewer inference steps. For example, when applied to SD1.5, the aesthetic score is first boosted from 5.26 to 5.54 without acceleration, and then further improved to 5.88 after being optimized by adversarial feedback learning. This demonstrates the superiority of our method in acceleration.

We also compared the two latest acceleration methods on SDXL, including the SDXL Turbo and SDXL Lightning. Although retaining the high text-to-image alignment, we found that the image generated by SDXL Turbo tends to lack fidelity, leading to an inferior FID score. SDXL Lightning achieves the most balanced performance in all of these aspects and reaches impressive aesthetic quality in 4-step inference. However, UniFL still obtains slightly better performance on these metrics.

**User Study.** We conducted a comprehensive user study using SDXL to evaluate the effectiveness of our method in enhancing generation quality and acceleration. As illustrated in Fig.3, our method significantly improves the original SDXL in terms of generation quality with a 68% preference rate and outperforms DreamShaper and DPO by 36% and 25% preference rate, re-

| Model | Step | FID↓ | CLIP Score↑ | Aes Score↑ |
|---|---|---|---|---|
| SD1.5-Base | 20 | 37.99 | 0.308 | 5.26 |
| SD1.5-IR [23] | 20 | 32.31 | 0.312 | 5.37 |
| SD1.5-DS [52] | 20 | 34.21 | 0.313 | 5.44 |
| SD1.5-DPO [22] | 20 | 32.83 | 0.308 | 5.22 |
| **SD1.5-UniFL** | 20 | **31.14** | **0.318** | **5.54** |
| SD1.5-Base | 4 | 42.91 | 0.279 | 5.16 |
| SD1.5-LCM [27] | 4 | 42.65 | 0.314 | 5.71 |
| SD1.5-DS LCM [26] | 4 | 35.48 | 0.314 | 5.58 |
| **SD1.5-UniFL** | 4 | **33.54** | **0.316** | **5.88** |
| SDXL-Base | 25 | 27.92 | 0.321 | 5.65 |
| SDXL-IR [23] | 25 | 26.71 | 0.319 | 5.81 |
| SDXL-DS [52] | 25 | 28.53 | 0.321 | 5.65 |
| SDXL-DPO [22] | 25 | 35.30 | 0.325 | 5.64 |
| **SDXL-UniFL** | 25 | **25.54** | **0.328** | **5.98** |
| SDXL-Base | 4 | 125.89 | 0.256 | 5.18 |
| SDXL-LCM [27] | 4 | 27.23 | 0.322 | 5.48 |
| SDXL-Turbo [24] | 4 | 30.43 | 0.325 | 5.60 |
| SDXL-Lighting [53] | 4 | 28.48 | 0.323 | 5.66 |
| **SDXL-UniFL** | 4 | **26.25** | **0.325** | **5.87** |

Table 1: **Quantitative comparison** between our method and other methods on SD1.5 and SDXL architecture. The best performance is highlighted with bold font, and the second-best is underlined.

spectively. Thanks to PeFL and decoupled aesthetic feedback learning, our method exhibits improvement even when compared to the competitive ImageReward, and is preferred by 17% additional people. In terms of acceleration, our method surpasses the widely used LCM by a substantial margin of 57% with 4-step inference. Even when compared to the latest acceleration methods like SDXL-Turbo and SDXL-Lightning, UniFL still demonstrates superiority and obtains more preference. This highlights the effectiveness of adversarial feedback learning in achieving acceleration.

**Qualitative Comparison.** As shown in Fig.4, UniFL achieves superior generation results compared with other methods. For example, when compared to ImageReward, UniFL generates images that exhibit a more coherent object structure (e.g., the horse), and a more captivating aesthetic quality (e.g., the cocktail). Notably, even with fewer inference steps, UniFL consistently showcases higher generation quality, outperforming other methods. It is worth noting that SDXL-Turbo, due to its modification of the diffusion hypothesis, tends to produce images with a distinct style.

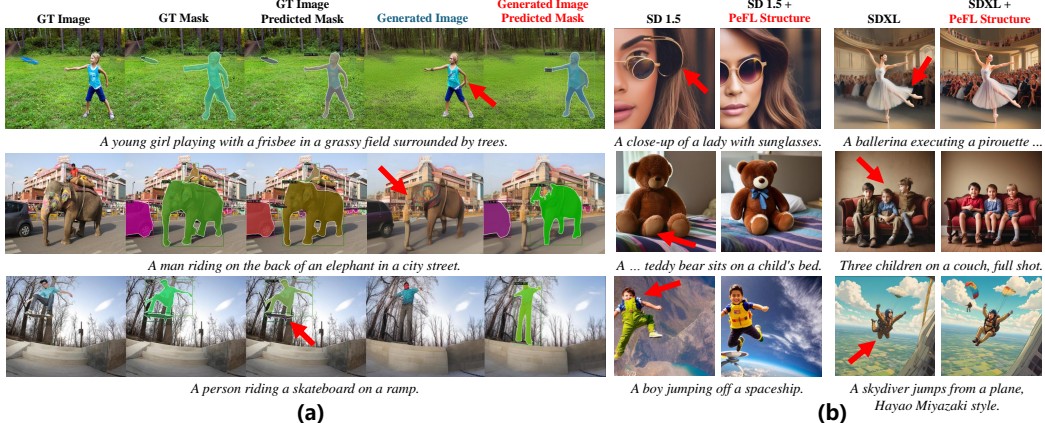

Figure 5: (a) Illustration of PeFL with instance segmentation model (SOLO). (b) Effect of PeFL on structure optimization.

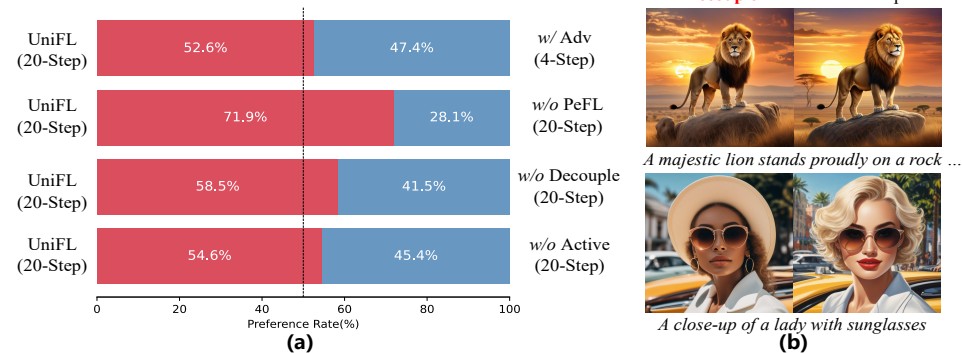

Figure 6: (a) Design components ablation of UniFL. (b) Visualization of decoupled and non-decoupled aesthetic feedback learning results.

## 5.3 Ablation Study

To validate the effectiveness of our design, we systematically remove one component at a time and conduct a user study. The results are summarized in Fig.6 (a). In the subsequent sections, we will further analyze each component. More results are presented in the Appendix.

**Superiority of PeFL.** As depicted in Fig.5 (a), PeFL leverages the instance segmentation model to capture the overall structure of the generated object effectively. By identifying structural defects, such as the distorted limbs of the little girl, the broken elephant, and the missing skateboard, PeFL provides more precise feedback signals for diffusion models. Such fine-grained flaws can not be recognized well with ReFL due to its global and coarse preference feedback, instead, the exploited professional visual perception provides more detailed and targeted feedback. As presented in Fig.5 (b), the PeFL significantly boosts the object structure generation (e.g. the woman's glasses, ballet dancer's legs). It is also demonstrated by the notable performance drop (71.9% vs 28.1%) when disabling the PeFL.

**Multiple Aspects Optimization with PeFL.** PeFL exploits various perceptual models to improve some particular visual aspects of the diffusion model and can easily be extended to multi-aspect optimization. As illustrated in Fig.8, the simultaneous incorporation of two distinct optimization objectives (style and structure optimization) does not compromise the effectiveness of each other. Take the prompt a baby Swan, graffiti as an example, integrating the style optimization via PeFL upon the base model successfully aligns the image with the target style. Further integrating the structure optimization objective preserves the intended style while enhancing the overall structural details (e.g. the feet of the Swan).

**Necessity of Decoupling Design.** We conducted an experiment that finetuned the SD1.5 using the same prompt set but a global aesthetic reward model trained with all dimensions' collected aesthetic preference data. As depicted in Fig.6 (b), the generated images are more harmonious and have an artistic atmosphere with the decoupled aesthetic reward tuning and are preferred by more 17%

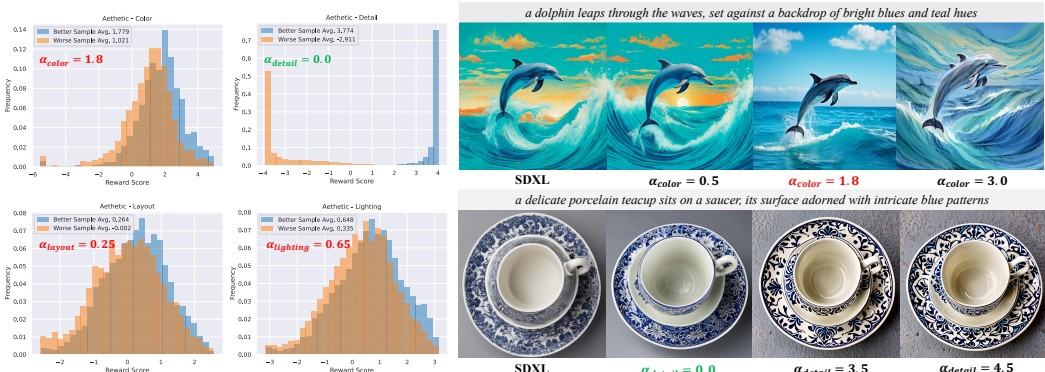

Figure 7: **Analysis on the** $\alpha_d$. Left: reward scores distribution on 5k validation preference image pairs with our final chosen values highlighted. Right: ablation on the $\alpha_d$ on *color* and *detail* reward.

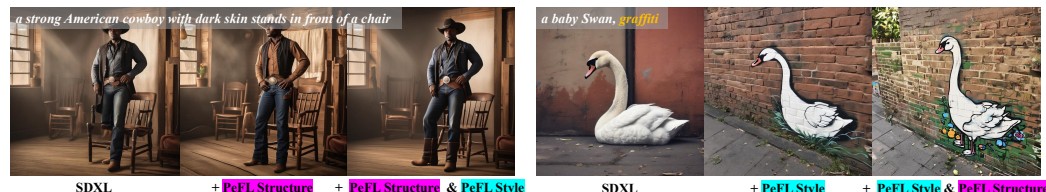

Figure 8: Incorporating the style and structure optimization objectives simultaneously with PeFL results in *no effectiveness degeneration of each other*.

individuals than the non-decoupled counterpart. This can be attributed to the ease of abstract aesthetic learning with the decoupling design. Moreover, it also can be found that aesthetic feedback learning with actively selected prompts leads to a higher preference rate (54.6% vs 45.4%) compared with the random prompts. Further analysis of the prompt selection can be found in the Appendix.B.2.

**Selection of Hinge Coefficient** $\alpha_d$**.** We select the hinge coefficient for each aesthetic reward model based on their reward distributions on the validation set. As illustrated in Fig.7 (left). there are clear margins in the reward scores between preferred and unpreferred samples. Moreover, such margin varies across these dimensions, emphasizing the necessity of the decoupled design. Empirically, we set $\alpha_d$ to the average reward scores of the preferred samples to encourage the diffusion model to prioritize generating samples with higher reward scores. Fig.7 (right) demonstrates that setting a small hinge for the "color" reward resulted in only minor improvement, while substantial coefficients led to image oversaturation. Optimal results were achieved by selecting a coefficient close to the average reward score of the preferred samples. A similar trend was observed for layout and lighting aesthetics from our experiments, except for the "detail" dimension. Interestingly, a slightly lower coefficient sufficed for satisfactory detail optimization, as a higher coefficient introduced more background noise. This could be attributed to the significant reward score difference between preferred and unpreferred samples, where a high coefficient could excessively guide the model toward the target reward dimension.

**Analysis on Adversarial Feedback Learning.** We analyzed the mechanism for the acceleration behind our adversarial feedback learning and found that (i) Adversarial training enables the reward model to provide guidance continuously. As shown in Fig.9 (a), the diffusion model offer suffers rapid overfitting when frozen reward models, known as reward hacks. By employing adversarial feedback learning, the trainable reward model (acting as the discriminator) can swiftly adapt to the distribution shift of the diffusion model output, significantly mitigating the over-optimization phenomenon, and allowing the reward to provide effective guidance for a longer duration. (ii) Adversarial training expands the time step of feedback learning optimization. The adversarial objective poses a strong constraint to force high-noise timesteps to generate clearer images, which allows the samples across all denoising steps to be rewarded properly. As presented in Fig.9 (b), when disabling the adversarial objective while retaining the full optimization timesteps during rewarding, the reward model fails to provide effective guidance for samples under fewer denoising steps due to the high-level noise, which

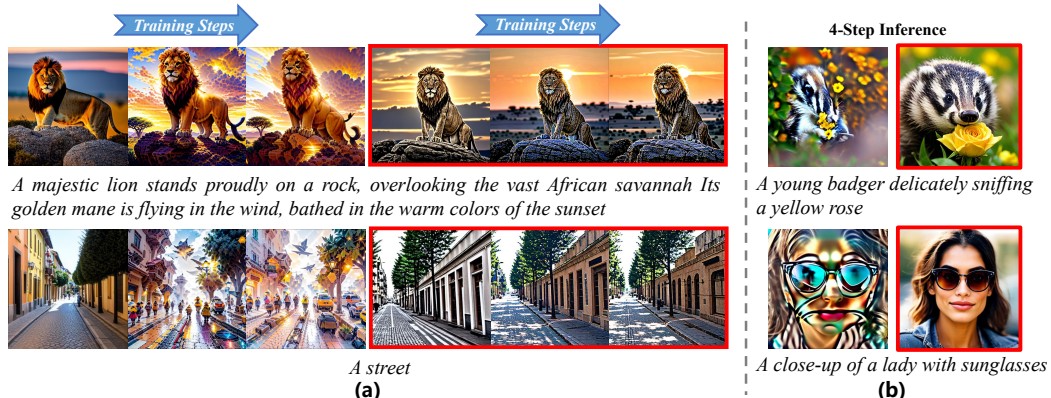

Figure 9: **Analysis of the benefits of adversarial training**. (a) It enables a longer optimization time for the reward model. (b) It enables the image under low denoising steps to be rewarded correctly. The red rectangle means incorporating the adversarial training objective.

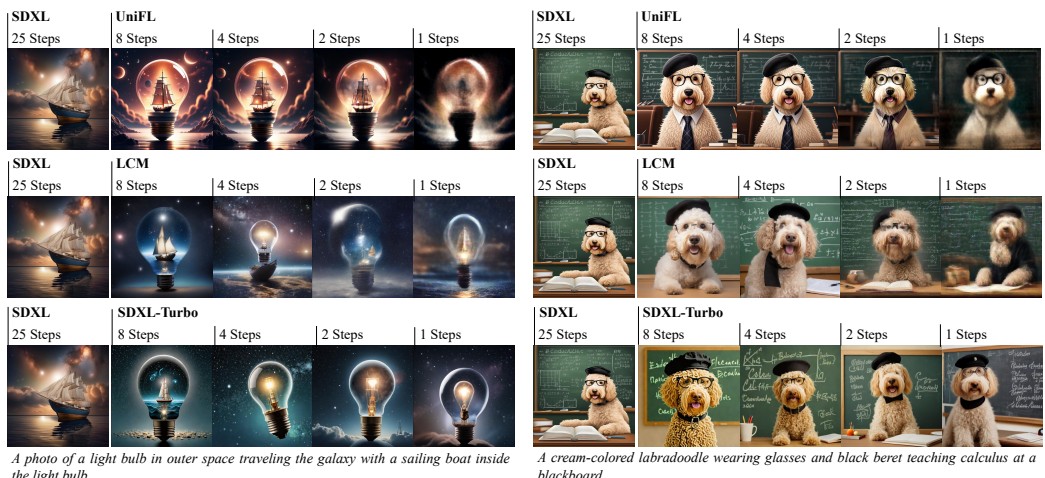

Figure 10: **Ablation on different inference steps** of UniFL.

results in poor generation results. With these two benefits, adversarial feedback learning significantly improves the generation quality of samples in lower inference steps and achieves superior acceleration performance ultimately. Notably, as shown in Fig.6 (a), the image generated with 4-step inference retains similar visual quality with 20-step inference (52.5% vs 47.4%) after going through the second stage training, which demonstrates the superiority of UniFL in acceleration.

**Ablation on Acceleration Steps.** We examine the acceleration capacity of UniFL under various inference steps, ranging from 1 to 8 as illustrated in Fig.10. Generally, UniFL performs exceptionally well with 2 to 8 inference steps with superior text-to-image alignment and higher aesthetic quality. The LCM method is prone to generate blurred images when using fewer inference steps and requires more steps (e.g., 8 steps) to produce satisfied images. However, both UniFL and LCM struggle to generate high-fidelity images with just 1-step inference, exhibiting a noticeable gap compared to SDXL-Turbo (e.g., the Labradoodle), which is intentionally designed and optimized for an extremely low-step inference regime. Therefore, there is still room for further exploration to enhance the acceleration capabilities of UniFL towards 1-step inference.

# 6 Conclusion

We propose UniFL, a framework that enhances visual quality, aesthetic appeal, and inference efficiency for latent diffusion models from the unified feedback learning perspective. By incorporating perceptual, decoupled, and adversarial feedback learning, UniFL can be applied to various latent diffusion models, such as SD1.5 and SDXL, and exceeds existing methods in terms of both generation quality enhancement and inference acceleration.

## Acknowledgements

This work was supported in part by the National Natural Science Foundation of China (NO. 62322608), in part by the CAAI-MindSpore Open Fund, developed on OpenI Community, in part by the Open Project Program of State Key Laboratory of Virtual Reality Technology and Systems, Beihang University (No.VRLAB2023A01).

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

# A Extend Details of Perceptual Feedback Learning

## A.1 Additional Examples of PeFL

The proposed perceptual feedback learning (PeFL) is highly flexible, allowing it to utilize different existing visual perception models to offer targeted visual quality feedback on particular aspects. To showcase the scalability of PeFL, we present two additional case studies where PeFL is employed to optimize style and layout generation.

i) **Style**: To effectively capture image style and provide feedback on style generation, we utilize the VGG-16 [56] model to encode image features and extract visual style using the well-established gram matrix in style transfer. Furthermore, we have curated a substantial dataset of approximately 150,000 high-quality artist-style text images. We leverage this dataset to conduct PeFL for style optimization. The objective of the optimization can be formulated as follows:

$$\mathcal{L}_{\text{pefl}}^{\text{style}}(\theta) = \mathbb{E}_{x_0 \sim \mathcal{D}, x_0' \sim G(x_{t_a})} \|\text{Gram}(V(x_0')) - \text{Gram}(V(x_0))\|_2, \tag{8}$$

where $V$ is the VGG network, and $\text{Gram}$ is the calculation of the gram matrix. We validate the effect of PeFL in style optimization based on SD1.5 and SDXL. Note that due to the newly introduced artist-style dataset, we compare our method with the DMs fine-tuned with the same style dataset via pre-train loss to ensure a fair comparison. As depicted in Fig.11, the PeFL significantly boosts style generation (e.g. 'frescos', 'impasto' style), enabling the model to generate the image with a more aligned style compared with applied pre-train loss (MSE loss). We further conduct the quantitive experiment to evaluate the effectiveness of PeFL on style optimization. Specifically, we collect 90 prompts about style generation and generate 8 images for each prompt. Then, we manually statistic the rate of correctly responded generation to calculate the style response rate. As shown in Tab.2, it is clear that the style PeFL greatly boosts the style generation on both architectures thanks to the superior style feedback provided by the VGG extracted feature, especially for SD1.5 with about 15% improvement. In contrast, leveraging naive diffusion pre-train loss for fine-tuning with the same collected style dataset suffers limited improvement due to stylistic abstraction missing in latent space.

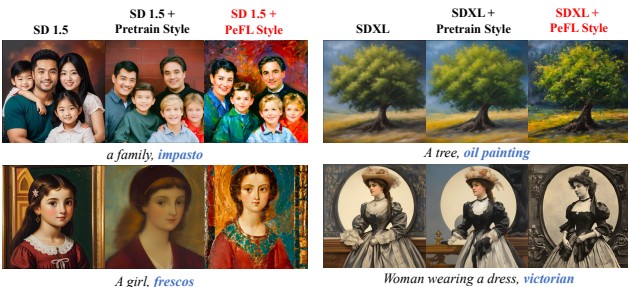

Figure 11: **Style optimization of PeFL** on SD1.5 and SDXL.

| Model | Style Response Rate |
|---|---|
| SD1.5 | 30.55% |
| SD1.5 + Style Pretrain | 35.25% |
| **SD1.5 + Style PeFL** | **45.14%** |
| SDXL | 66.67% |
| SDXL + Style Pretrain | 68.34 % |
| **SDXL + Style PeFL** | **75.27%** |

Table 2: **Quantitive performance of PeFL** in style generation.

ii) **Layout**: Generally, the semantic segmentation map characterizes the overall layout of the image as shown in Fig.12 (a). Therefore, semantic segmentation models can serve as a better layout feedback provider. Specifically, we utilize the visual semantic segmentation model to execute semantic segmentation on the denoised image $x_0'$ to capture the current generated layout and supervise it with the ground truth segmentation mask and calculate semantic segmentation loss as the feedback on the layout generation:

$$\mathcal{L}_{\text{pefl}}^{\text{layout}}(\theta) = \mathbb{E}_{x_0 \sim \mathcal{D}, x_0' \sim G(x_{t_a})} \mathcal{L}_{\text{semantic}}(m_s(x_0'), \text{GT}(x_0)) \tag{9}$$

where $m_s$ represents the semantic segmentation model, $\text{GT}(x_0)$ is the ground truth semantic segmentation annotation and the $\mathcal{L}_{\text{semantic}}$ is the semantic segmentation loss depending on the specific semantic segmentation model. We conducted an experiment on PeFL layout optimization based on SD1.5. Specifically, we utilize the COCO Stuff [57] with semantic segmentation annotation as the semantic layout dataset and DeepLab-V3 [58] as the semantic segmentation model. The results are presented in Fig.12 (b). It demonstrates that the PeFL significantly improves the layout of the generated image, for instance, the bear on the bed in a diagonal layout. Note that here we

focus on the objective layout generation that is explicitly mentioned in the prompts, for example, 'stands', 'overlooking', and 'sit at'. As a comparison, the layout reward model used in aesthetic feedback learning primarily emphasizes the subjective composition from the aesthetic angle, which may not be described clearly by the textual prompt. We further conduct the user study to evaluate the effectiveness of PeFL with the semantic segmentation model quantitatively. As shown in Fig.13 (b), we are surprised to find that the image details also observed a significant boost in addition to the improvement in the layout generation. This probably stems from the dense per-pixel feedback from the semantic segmentation objective.

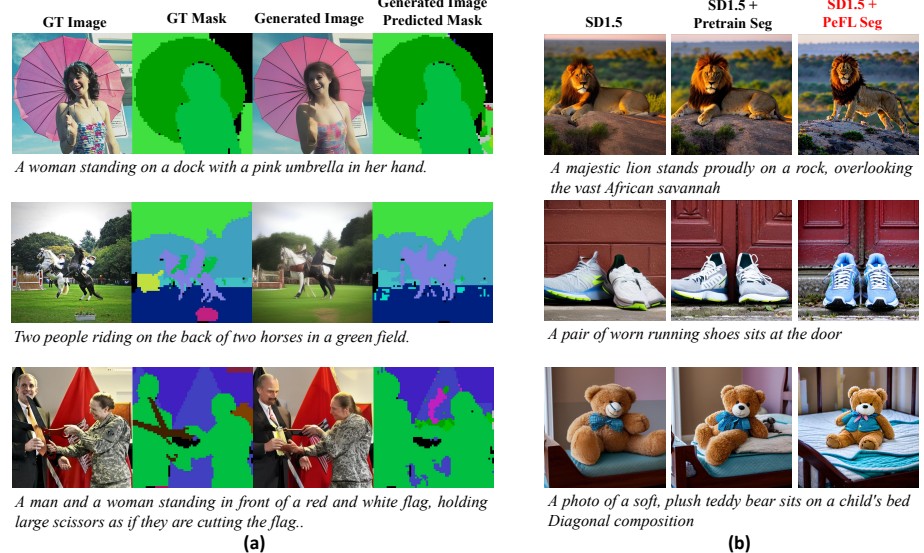

Figure 12: (a) The illustration of the PeFL on the layout optimization. The semantic segmentation model captures the layout and text-to-image misalignment between the ground truth image and the generated image (DeepLab-V3 [58] is taken as the segmentation model). (b) The layout optimization effect of the PeFL with semantic segmentation model on SD1.5.

Indeed, PeFL is an incredibly versatile framework that can exploit a wide range of visual perceptual models, such as OCR models [59, 60] and edge detection models [61, 62], to boost the performance of LDMs. Furthermore, we are actively delving into utilizing the visual foundation model, such as SAM [63], which holds promising potential in addressing various visual limitations observed in current diffusion models.

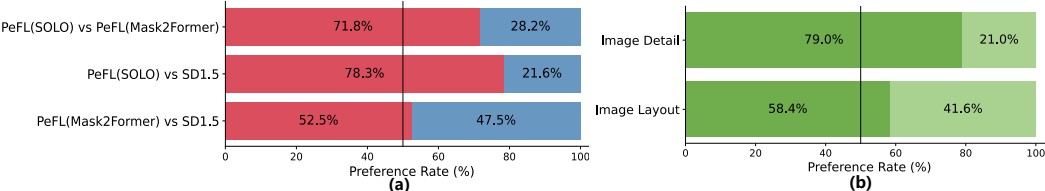

Figure 13: (a) The user study results on the ablation of different instance segmentation models in PeFL during structure optimization. PeFL (SOLO): PeFL fine-tune SD1.5 with SOLO as the instance segmentation model. PeFL (Mask2Former): PeFL fine-tune SD1.5 with Mask2Former as the instance segmentation model. (b) The user study results on the effect of PeFL layout optimization. Dark Green: SD1.5 with PeFL, Light Green: SD1.5 without PeFL.

## A.2 Ablation on Visual Perceptual Model

PeFL utilizes various visual perceptual models to provide visual feedback in specific dimensions to improve the visual generation quality on particular aspects. Different visual perceptual models of a certain dimension may have different impacts on the performance of PeFL. Taking the structure optimization of PeFL as an example, we investigated the impact of the accuracy of instance segmentation models on PeFL performance. Naturally, the higher the precision of the instance segmentation,

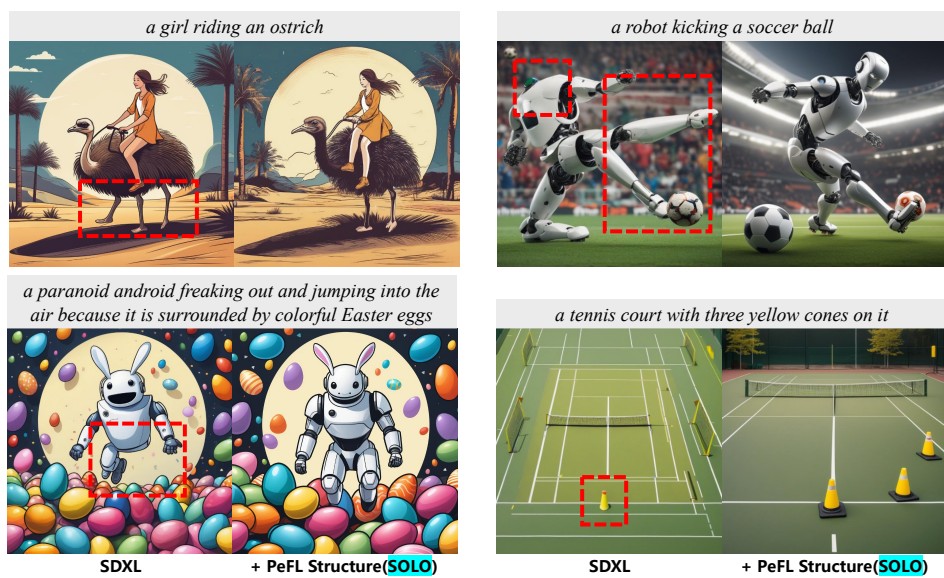

Figure 14: **Generalization of PeFL with SOLO**. The generation of the concepts not included in COCO (e.g. ostrich, robot, cones) is also improved after PeFL optimization.

the better the performance of structure optimization. To this end, we choose the Mask2Former [64], another representative instance segmentation model with state-of-the-art performance to achieve structure optimization with PeFL. The results are shown in Fig.16 (a) and Fig.13 (a). It is intriguing to note that the utilization of a higher precision instance segmentation model does not yield significantly improved results in terms of performance. We speculate it lies in the different architectures of the instance segmentation of these two models. In SOLO [50], the instance segmentation is formulated as a pixel-wise classification, where each pixel will be responsible for a particular instance or the background. Such dense supervision fashion enables the feedback signal to better cover the whole image during generation. In contrast, Mask2Former [64] takes the query-based instance segmentation paradigm, where only a sparse query is used to aggregate the instance-related feature and execute segmentation. This sparse nature of the query-based method makes the feedback insufficient and leads to inferior fine-tuning results. We leave further exploration of how to choose the most appropriate visual perceptual model for feedback tuning to future work.

### A.3 Generalization of PeFL with Close-set Perceptual models

We utilize the SOLO instance segmentation model trained with close-set dataset COCO for PeFL structure optimization. One may be concerned that this will lead to limited concepts that PeFL can optimize (i.e. only the COCO concept). However, on the one hand, although we apply the SOLO instance segmentation model trained on the COCO dataset (80 categories), we observe that PeFL exhibits exceptional generalization capability and the generation performance of many concepts not shown in the COCO dataset is also boosted significantly as shown in Fig.14. We believe LDM can be guided to learn general and reasonable structure generation via PeFL optimization. On the other hand, the proposed perceptual feedback learning is a very flexible framework, and it is very straightforward to replace the close-set model SOLO with other open-set instance segmentation models such as Ground-SAM to achieve further improvement for the concepts in the wild(e.g. concepts in LAION dataset).

## B  Extend Details of Decoupled Feedback Learning

### B.1  Aesthetic Preference Data Collection

We break down the general and coarse aesthetic concept into more specific dimensions including color, layout, detail, and lighting to ease the challenge of aesthetic fine-tuning. We then collect the human preference dataset along each dimension. Specifically, we employ the SDXL [40] as the base model and utilize the prompts from the MidJourney [3, 65] as input, generating two images for

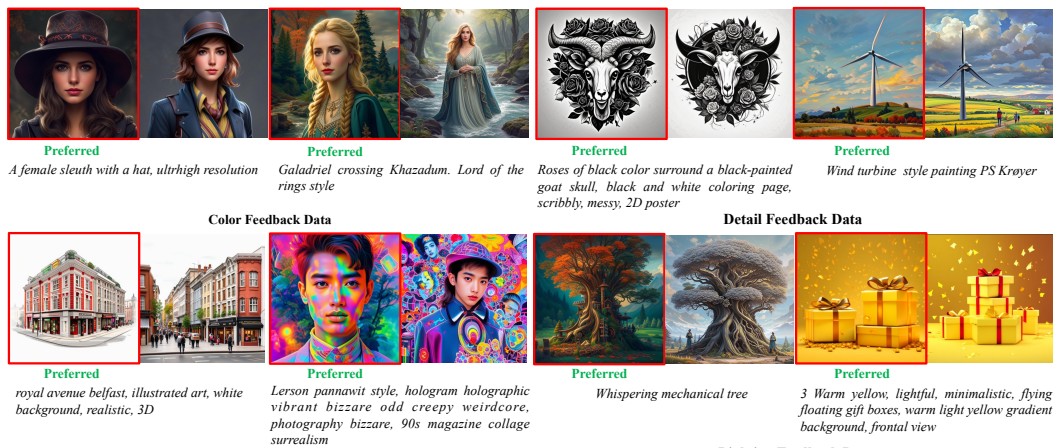

**Color Feedback Data**

*A female sleuth with a hat, ultrhigh resolution*

*Galadriel crossing Khazadum. Lord of the rings style*

*Roses of black color surround a black-painted goat skull, black and white coloring page, scribbly, messy, 2D poster*

**Detail Feedback Data**

*Wind turbine style painting PS Krøyer*

*royal avenue belfast, illustrated art, white background, realistic, 3D*

**Layout Feedback Data**

*Lerson pannawit style, hologram holographic vibrant bizzare odd creepy weirdcore, photography bizzare, 90s magazine collage surrealism*

*Whispering mechanical tree*

**Lighting Feedback Data**

*3 Warm yellow, lightful, minimalistic, flying floating gift boxes, warm light yellow gradient background, frontal view*

Figure 15: Decoupled aesthetic feedback data examples. The preferred samples are highlighted with red rectangles.

each prompt. Subsequently, we enlist the expertise of 4 to 5 professional annotators to assess and determine the superior image among the generated pair. Given the inherently subjective nature of the judgment process, we have adopted a voting approach to ascertain the final preference results for each prompt. Finally, we curate 30,000, 32,000, 30,000, and 30,000 data pairs for the color, layout, detail, and lighting dimensions, respectively. Examples of the collected aesthetic feedback data of different dimensions are visually presented in Fig.15.

## B.2 Active Prompt Selection

**Prompt Selection Process.** We introduce an active prompt selection strategy designed to choose the most informative and diverse prompts from a vast prompt database. The comprehensive implementation of this selection strategy is outlined in the Algorithm.2. Our strategy's primary objective is to select prompts that offer maximum information and diversity. To accomplish this, we have devised two key components: the *Semantic-based Prompt Filter* and the *Nearest Neighbor Prompt Compression*. The semantic-based prompt filter is designed to assess the semantic relationship embedded within the prompts and eliminate prompts that lack substantial information. To accomplish this, we utilize an existing scene graph parser[2] as a tool to parse the grammatical components, such as the subjective and objective elements. The scene graph parser also generates various relationships associated with the subjective and objective, including attributes and actions. We then calculate the number of relationships for each subjective and objective and select the maximum number of relationships as the measurement of the information amount encoded in the prompt. A higher number of relationships indicates that the prompt contains more information. We filter out prompts that have fewer than $\tau_1 = 1$ relationships, which discard the meaningless prompt like 'ff 0 0 0 0' to reduce the noise of the prompt set. Upon completing the filtration process, our next objective is to select a predetermined number of prompts that exhibit maximum diversity. To achieve this, we adopt an iterative process to achieve this objective. In each iteration, we randomly select a seed prompt and subsequently suppress its nearest neighbor[3] prompts that have a similarity greater than $\tau_2 = 0.8$ as illustrated in Fig.16 (b). The next iteration commences with the remaining prompts, and we repeat this process until the similarity between the nearest neighbors of all prompts falls below the threshold $\tau_2$. Finally, we randomly select the prompts, adhering to the fixed number required for preference fine-tuning.

**Analysis of the Actively Selected Prompts.** As illustrated in Fig.6 (a) in the main paper, our strategically chosen prompts yield superior performance in aesthetic feedback learning. To further comprehend the advantage of this design, we present the training loss curve in Fig.16 (c), comparing the use of actively selected prompts versus random prompts. It clearly shows that the diffusion model rapidly overfits the guidance provided by the reward model when using the randomly selected prompts, ultimately resulting in the loss of effectiveness of the reward model quickly. One contributing

---

[2] https://github.com/vacancy/SceneGraphParser
[3] https://github.com/facebookresearch/faiss

---

**Algorithm 2** Active prompt selection for decoupled aesthetic feedback learning

---

**Input:** Initial prompt database: $\mathcal{D}$; number of desired prompts: $N$; $\tau_1$ and $\tau_2$: relation and similarity threshold.
**Output:** The selected prompt set: $\mathcal{SP}$
1: $\mathcal{P} = \varnothing$
2: *# Semantic-based Prompt Filter*
3: **for** $p_i \in \mathcal{D}$ **do**
4:    $\mathcal{SR} \leftarrow$ SemanticParser($p_i$)
5:    **if** $|\mathcal{SR}| > \tau_1$ **then**
6:      $\mathcal{P} \leftarrow p_i$ // choose informative prompt
7:    **end if**
8: **end for**
9: *# Nearest Neighbor Prompt Compression*
10: $I \leftarrow$ shuffle(range(len($|\mathcal{P}|$)))
11: $R \leftarrow$ False // set the removed prompt array
12: $S \leftarrow \varnothing$ // set the selected prompt index
13: Dist, Inds $\leftarrow$ KNN($R$, $k$) // K-nearest neighbor of each prompt
14: **for** index $I_i \in I$ **do**
15:    **if** not $R[I_i]$ and $I_i$ not in $S$ **then**
16:      $S \leftarrow I_i$ // append the selected prompt
17:      dist, inds = Dists[$I_i$], Inds[$I_i$] // K-nearest neighbor similarities
18:      **for** index $d_i \in$ inds **do**
19:        **if** dist[$d_i$] > $\tau_2$ **then**
20:          $R[d_i]$ = True
21:        **end if**
22:      **end for**
23:    **end if**
24: **end for**
25: $\mathcal{SP} \leftarrow$ RandomSelect($\mathcal{P}$, $S$, $N$) // randomly select $N$ diverse prompts according the retained index
26: **return** $\mathcal{SP}$

---

factor to this phenomenon is the distribution of prompts for optimization. If the prompts are too closely distributed, the reward model is forced to frequently provide reward signals on similar data points, leading to the diffusion model rapidly overfitting and collapsing within a limited number of optimization steps. We statistic the average nearest embedding similarity (ANS) of the 100K prompts randomly selected from DiffusionDB [48] by calculating the cosine similarity between each prompt with its most similar prompt within the embedding space and taking the average over all the prompts. The ANS of the randomly selected prompts is approximately 0.89, which delivery highly redundant. As a comparison, the prompts selected by our strategy exhibit considerable diversity with ANS of 0.73, enabling a more balanced and broad reward calculation, which eases the over-fit significantly. Therefore, with the actively selected prompts, the diffusion model obtains a more comprehensive feedback signal and can be optimized toward human preference more efficiently.

## C   Generalization Study

To further verify the generalization of UniFL, we performed downstream tasks including LoRA, and ControlNet Specifically, we selected several popular styles of LoRAs [7], and several types of ControlNet [9] and inserted them into our models respectively to perform corresponding tasks. As shown in Fig.17, our model demonstrates excellent capabilities in style adaptation and controllable generation.

## D   More Visualization Results

We present more visual comparison between different methods in Fig.18. It demonstrates the superiority of UniFL in both the generation quality and the acceleration. In terms of generation quality, UniFL exhibits more details (e.g. the hands of the chimpanzee), more complete structure (e.g. the dragon), and more aesthetic generation (e.g. the baby sloth and the giraffe) compared with DPO and ImageReward. In terms of acceleration, the LCM tends to generate a blurred image, while the SDXL-Turbo generates the image with an unpreferred style and layout. As a comparison, UniFL still retains the high aesthetic detail and structure under the 4-step inference.

## E   Discussion and Limitations

UniFL demonstrates promising results in generating high-quality images. However, there are several avenues for further improvement:

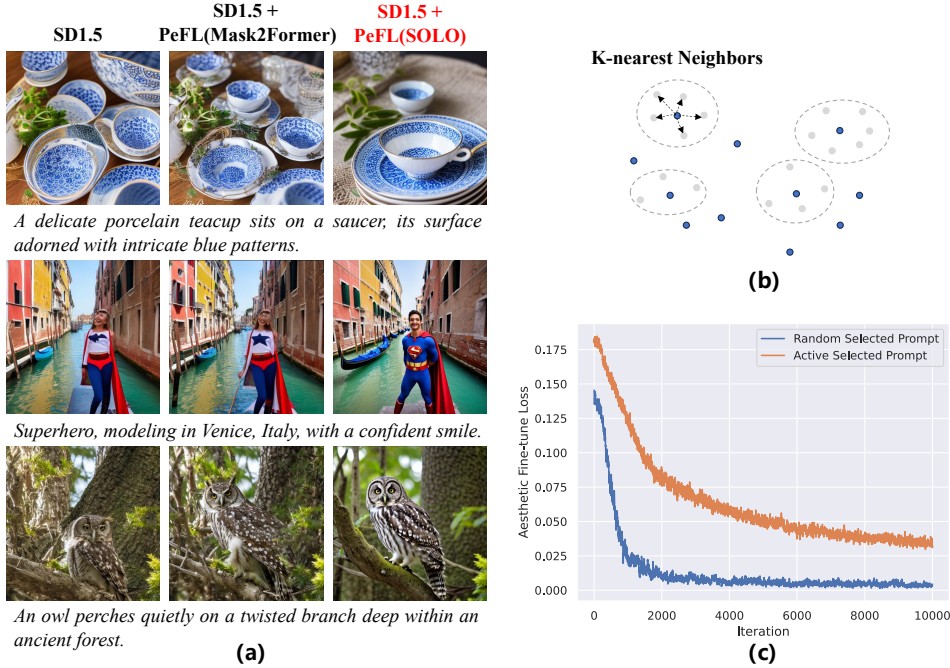

Figure 16: (a) The visual comparison between the PeFL structure optimization with different instance segmentation models. (b) Illustration of Nearest Neighbor Prompt Compression. (c) Training loss curve when utilizing different prompts for decoupled aesthetic feedback learning.

**Large Visual Perception Models**: We are actively investigating the utilization of advanced large visual perception models to provide enhanced supervision.

**Extreme Acceleration**: While the current 1-step model's performance may be relatively subpar, the notable success we have achieved in 4-step inference suggests that UniFL holds significant potential for exploration in one-step inference.

**Streamlining into a Single-stage Optimization**: Exploring the possibility of simplifying our current two-stage optimization process into a more streamlined single-stage approach is a promising direction for further investigation.

# F Broader Impact

The proposed framework, UniFL, has the potential to have significant broader impacts in the field of image generation and related downstream applications. The improved visual quality of image generation achieved through UniFL can enhance various applications that rely on generated images, including computer graphics, virtual reality, and content creation. This can lead to the creation of more realistic and visually appealing virtual environments, improved visual effects in movies and video games, and better-quality generated content for digital media. However, it is also important to consider the potential ethical implications and societal impacts of advancements in image generation techniques. With the ability to generate highly realistic images, there is a risk of misuse or abuse, such as the creation of deepfake content for malicious purposes. Researchers, developers, and policymakers must be vigilant and consider the ethical implications of these advancements, promoting responsible use and raising awareness about the potential risks associated with synthetic media.

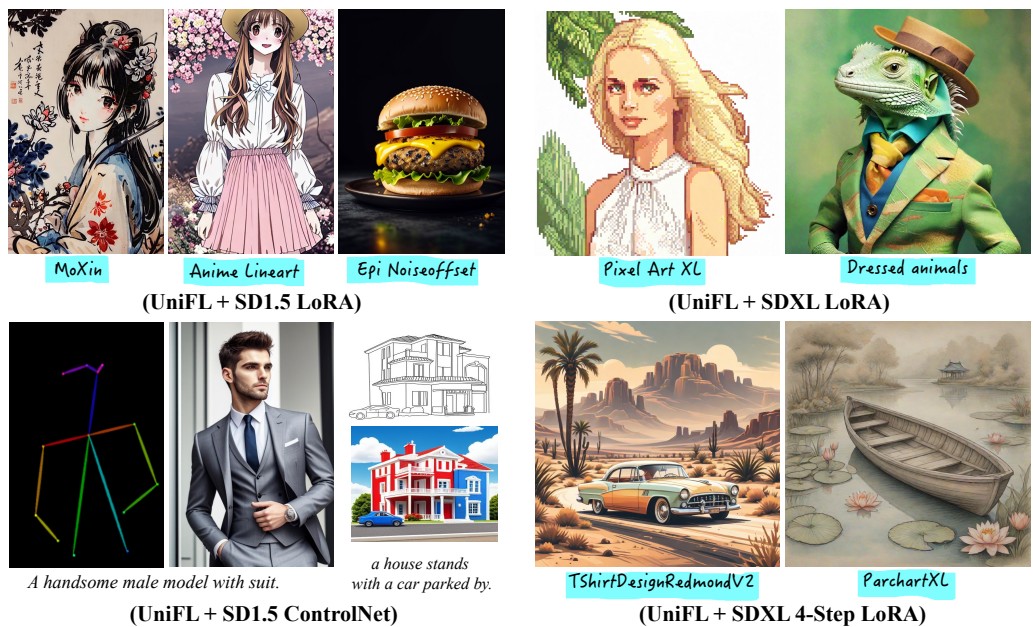

**(UniFL + SD1.5 LoRA)**      **(UniFL + SDXL LoRA)**

*A handsome male model with suit.*

*a house stands with a car parked by.*

**(UniFL + SD1.5 ControlNet)**      **(UniFL + SDXL 4-Step LoRA)**

Figure 17: Both SD1.5 and SDXL still keep high adaptation ability after being enhanced by the UniFL, even after being accelerated and inference with fewer denoising steps.

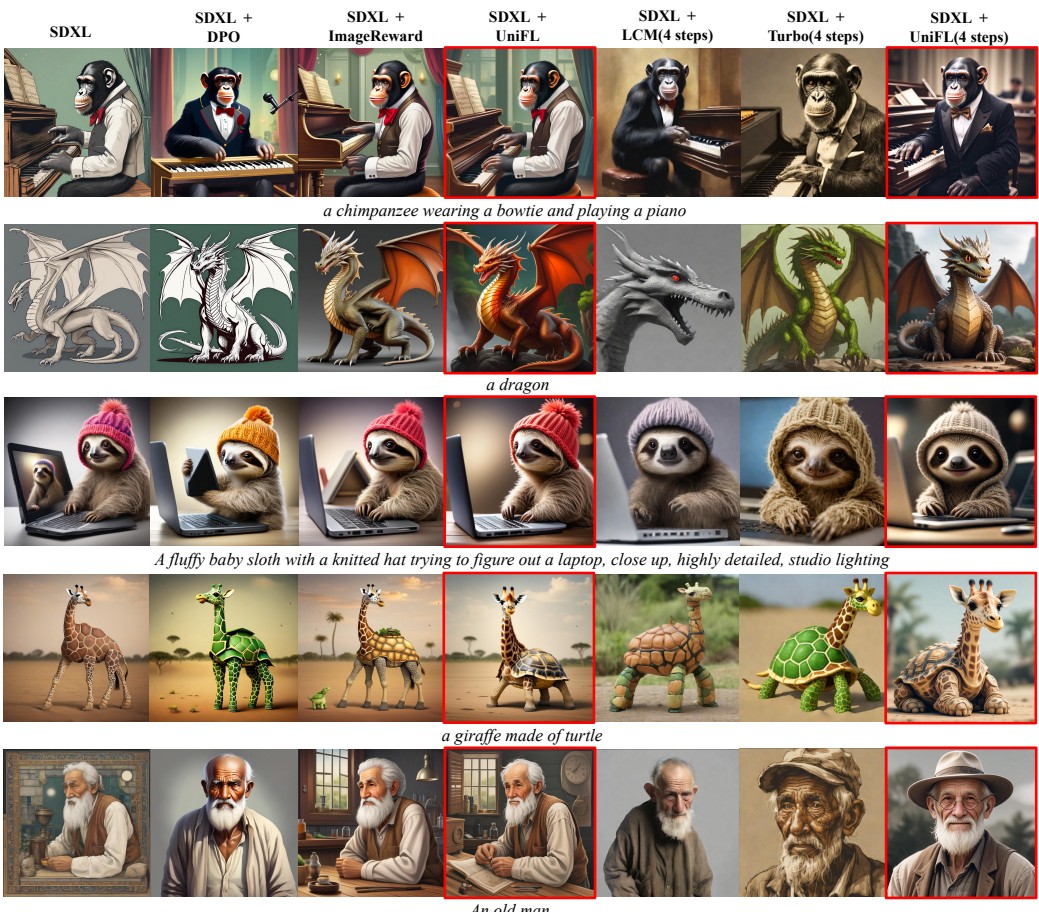

Figure 18: More visual comparison with different methods. UniFL yields better prompt-aligned and visually appealing results than other methods. Our results are highlighted with red rectangles.

