# OpenReview forum: "UniFL: Improve Latent Diffusion Model via Unified Feedback Learning"
_NeurIPS.cc/2024/Conference — NeurIPS 2024 poster_

### Official Review · Reviewer_af1S · 2024-06-18

**Soundness:** 3
**Presentation:** 3
**Contribution:** 3
**Rating:** 6
**Confidence:** 2

**Summary:**

This paper introduces UniFL, a novel approach for improving diffusion models through unified feedback learning. The objective of UniFL is to enhance visual generation quality, preference aesthetics, and inference acceleration. To achieve this, the paper proposes three key components: perceptual feedback learning, decoupled aesthetic feedback learning, and adversarial feedback learning. UniFL is designed to be a two-stage training pipeline that can be applied to various models and yields impressive improvements in both generation quality and acceleration. The paper provides experimental results demonstrating the effectiveness of UniFL in terms of generation quality and acceleration. Additionally, the paper discusses the potential broader impacts and ethical implications of advancements in image generation techniques.

**Strengths:**

1. The writing of this article is commendable, with a well-structured format.
2. The author conducted a large number of visualization experiments to illustrate the focus and effectiveness of the method, which is very attractive.
3. The proposed method serves as a plug-and-play indeed improves the performance on both SD15 and SDXL.

**Weaknesses:**

1. There are some minor LaTeX formatting issues: for example, there should be a consistent space before parentheses in abbreviations, the quotes in the Appendix should be implemented using `’, and “our” should be “Our” in line 130.
2. The values of hyper-parameters such as \alpha_d are not explicitly stated in the paper.

**Questions:**

1. Have the authors explored the impact of different aesthetics on generated images?
2. Are the user studies enough to prove the effectiveness of each module in this field?
3. Why not compare with SDXL-IR and SDXL-DPO in Ablation on Acceleration Steps?

**Limitations:**

Yes.

---

> ### Author Rebuttal · Authors · 2024-08-06
>
> Thanks for your kind words about our good writing, sufficient experiments, and the effectiveness of our method. We would like to answer the proposed questions in the following:
>
> 1. **Minor Typos**:  Thanks for the suggestion, we will modify these places in our revised version.
> 2. **Selection of hinge coefficient $\alpha_d$**: Please refer to the Author Rebuttal part and Fig.4 of the global rebuttal PDF for more details.
> 3. **Impact of different aesthetics during generation**: The impact of different aesthetic reward models in the generated images can be clearly observed in Fig.3 in our main paper. As can be seen, with the integration of color aesthetic rewards, the colors of the images generated by UniFL are more harmonious and natural. This enhancement is particularly pronounced in the acceleration scenario, where LCM and Turbo exhibit color deterioration characterized by darker, grayer, and vaguer, contrasting with our method's maintenance of a lively color palette. The effects of the detail and layout reward model are also notable. For instance, when considering the prompt `A bloody may cocktail`,  the image generated by UniFL showcases intricate background details while maintaining a well-balanced layout with the foreground, resulting in a visually appealing composition. Furthermore, the lighting aesthetic reward ensures the lighting with an atmosphere in the image, as opposed to the flat center background lighting observed in alternative methods.
> 4. **Effectiveness of User Study**: Due to the subjective nature of the image quality judgment, the most effective way to evaluate the performance of a model in the context of text-to-image generation so far is still the user evaluation. This is also the standard practice of various classical text-to-image works such as SDXL[1], SDXL-Turbo[2], etc. Moreover, to increase reliability, we involve a considerable number of users(10 users) in our quality evaluation study, which we believe can validate our conclusion relatively accurately.
> 5. **Acceleration ablation with DPO/ImageReward**: We did not choose to compare the accelerated version of our method with the SDXL-IR and SDXL-DPO as _they are not the methods tailored for inference acceleration instead of generation quality optimization_. There is no few-steps optimized version of SDXL of ImageReward and DPO for fair comparison with our method of inference acceleration. Even though, the unaccelerated version SDXL optimized via our approach still displays more superior image quality than these methods as evidenced by our experiments.
>
> We hope our rebuttal can address your concerns.
>
> [1] SDXL: Improving Latent Diffusion Models for High-Resolution Image Synthesis
>
> [2] SDXL-Turbo: Adversarial Diffusion Distillation

---

> > ### Comment · Reviewer_af1S · 2024-08-12
> >
> > Thank you for your response. I have checked the rebuttal and tend to keep my score.

---

### Official Review · Reviewer_dHXQ · 2024-07-05

**Soundness:** 3
**Presentation:** 3
**Contribution:** 3
**Rating:** 6
**Confidence:** 5

**Summary:**

Considering that current diffusion models still suffer from several limitations, this paper aims to propose a unified framework, UniFL to address the main existing challenges by applying feedback learning. To respectively solve the issues of visual distortion, poor aesthetic appeal, and inefficient inference, this paper demonstrates different sub-modules, namely Perceptual Feedback Learning, Decoupled Feedback Learning, and Adversarial Feedback Learning. By fully leveraging the ability of Perceptual Feedback Learning to fine-tune the diffusion during the two training stages, the proposed model aims to achieve both remarkable generation quality and speed simultaneously. Sufficient experiments, including extensive ablation studies for different proposed sub-modules, have proved the validation and effectiveness of the model’s design.

**Strengths:**

The proposed idea about a combination of feedback learning and diffusion models is well introduced with necessary background information and is sufficiently motivated.

The three main issues remaining for diffusion, namely inferior quality, lack of aesthetics, and slow inference speed are well addressed, keeping the following methodology sections’ logic pipeline clear and understandable.

The proof of the methodology part is adequately explained with abundant support of pipeline figures and pseudocode that help the reader get full access to the novelty insight.

Sufficient experiments over different domains, especially the wide range of ablation studies, have been carried out validly which demonstrate the effectiveness of all the proposed feedback learning modules, which enhances persuasiveness.

**Weaknesses:**

The proposed qualitative visualization comparison mainly focuses on justifying the overall generation style and the structure correctness. However, a main issue that may occur during the inference speed-up is detail loss. It will be more appreciated if the author can provide some further visual comparison results to demonstrate the model’s ability to keep the visualization results consistent with that adjective in the text prompt context.

Also, according to Figure 7, an unwelcomed issue can be found that the images generated by UniFL for text prompt A cream-colored labradoodle wearing glasses and a black beret teaching calculus at a blackboard mistakenly wears a black tie, which has never been mentioned in the text prompt. It shows an unexpected trend of overfitting and inconsistency. Further explanation about the cause of such results should be addressed.

**Questions:**

Please see the weakness part.

**Limitations:**

Though there are still some blemishes part for some of the proposed results, generally the experiment part is well-organized, and filled with abundant ablation studies for the proposed modules.

---

> ### Author Rebuttal · Authors · 2024-08-06
>
> Thanks for your kind words about our well-motivated method, sufficient experiments, and good writing. We would like to answer the proposed questions in the following:
> 1. **Visualization on T2I alignment after acceleration**:  We visualize the text-to-image(T2I) alignment performance of SDXL accelerated via our method on the widely used benchmark prompt set - DrawBench, and provide some results in Fig.3 of the global rebuttal PDF. It clearly demonstrates that, even with a reduced number of inference steps, our approach maintains superior text-image consistency under various circumstances after acceleration, including attribute binding, object counting, and counterfactual prompt inputs.
> 2. **Explanation of the unexpected generation**: It should be noted that although it is not what the user expected,  there is nothing wrong with the model to generate this black tie, as the input prompt does not explicitly specify that a tie is undesired and all the stuff mentioned in the prompt(e.g. cream-colored labradoodle, glasses, black beret. etc) has already correctly generated. In other words, it is free for the model to generate additional items in addition to the input prompt specified. One possible reason for the model to generate the black tie in this case is the data bias residing in the training samples. That is, there are vast training examples of teachers lecturing at a blackboard, where most teachers in these examples all wear ties. Such co-occurrence bias makes the model easily generate an extra tie when comes to the character in front of the blackboard.  A possible solution is to input an additional negative prompt to suppress such undesired behavior.
>
> We hope our rebuttal can address your concerns.

---

> > ### Comment · Reviewer_dHXQ · 2024-08-13
> >
> > Thank you for providing the response. After reading it, I have decided to keep the original score.

---

### Official Review · Reviewer_nwEj · 2024-07-13

**Soundness:** 3
**Presentation:** 3
**Contribution:** 2
**Rating:** 5
**Confidence:** 4

**Summary:**

The work introduces Unified Feedback Learning (UniFL), a unified framework to enhance diffusion models through feedback learning. It addresses three main challenges in diffusion models: visual quality, aesthetic appeal, and inference efficiency. UniFL comprises perceptual feedback learning, decoupled feedback learning, and adversarial feedback learning.
- Perceptual Feedback Learning utilizes existing perceptual models, such as an instance segmentation model, to fine-tune diffusion models on a specific aspect.
- Decoupled Feedback Learning decomposes the general aesthetic concept into color, layout, lighting, and detail. It fine-tunes the model by reward feedback learning in all these sub-categories.
- Adversarial Feedback Learning exploits a general reward model as a discriminator to improve the generation quality of fewer denoising steps.
Experiments demonstrate that UniFL improves the performance of diffusion models like SD1.5 and SDXL in terms of generation quality, aesthetic preference, and inference speed.

**Strengths:**

- The paper is clear and easy to follow
- The proposed method utilizes different priors from other perceptual models to improve diffusion models.
- New aesthetic human feedback dataset is proposed.

**Weaknesses:**

- The overall method is some modification from ReFL.
- The perceptual feedback learning part is limited by the selected models, which seem to only fine-tune concepts used to train these perceptual models.
- The performance is not significantly superior to other methods according to Table 1.

**Questions:**

- If users want to generate images containing concepts not shown in COCO dataset, but shown in LAION, it seems perceptual feedback learning part will not work.
- If users want to fine-tune the diffusion model in several different perceptual aspects simultaneously, will the priors from different perceptual models interrupt each other?
- Weaknesses above

**Limitations:**

Yes, the authors addressed the limitations.

---

> ### Author Rebuttal · Authors · 2024-08-06
>
> Thanks for your kind words about our writing and contribution to the feedback dataset curation. We would like to answer the proposed questions in the following:
>
> 1. **Comparison with ReFL**: There are significant differences between our method and ReFL. Specifically: 1) Given the input prompt, ReFL starts with pure noise to obtain the denoised image and then imposes the reward scoring. This limits their usage to the reward models pre-trained on the collected preference data for the reward guidance. In contrast, _we incorporate an extra image condition for noise initialization and use the perceptual label of this condition image for feedback supervision on the denoised image_. Such formulation allows us to leverage the prior knowledge of a wide range of existing perceptual models to perform feedback fine-tuning. 2) In ReFL, a single coarse-grained preference reward model is applied, which may encounter the rewarding conflict, resulting in insufficient fine-tuning. As a comparison,  _we develop multiple decoupled aesthetic reward models, which enable more effective fine-grained aesthetic preference alignment_. 3) A core finding exhibited in ReFL is that it can only be applied to later denoising time steps during fine-tuning with the reward model as the reward model cannot reward correctly in the early steps. By contrast,  _we introduce adversarial feedback supervision, which effectively removes this limitation on optimization steps, allowing us to apply feedback fine-tuning to any denoised time step_, thereby achieving model inference acceleration. Overall, these novel designs make our approach distinct from ReFL and yield a more flexible and effective feedback fine-tuning framework for LDM.
> 2. **Fine-tune concepts limitation in PeFL**:  On the one hand, although we apply the SOLO instance segmentation model trained on the COCO dataset(80 categories), we observe that PeFL exhibits exceptional generalization capability and the generation performance of many concepts not shown in the COCO dataset is also boosted significantly as shown in Fig.2 of the global rebuttal PDF. We believe that the LDM can be guided to learn general and reasonable structure generation via PeFL optimization. On the other hand, the proposed perceptual feedback learning is a very general module, and it is very straightforward to replace the close-set model SOLO with other open-set instance segmentation models such as Ground-SAM to achieve further improvement for the concepts in the wild(e.g. concepts in LAION dataset).
> 3. **About the performance in Tab.1**: We argue that our method demonstrates notable improvements compared to other approaches as outlined in Tab.1.  It is essential to emphasize that different metrics possess unique properties, and _absolute enhancement should not be the sole criterion for evaluation_. For instance, the inherent limitations in the fine text-to-image alignment of the CLIP model make it challenging to achieve significant advancements in CLIP scores as it cannot capture the nuanced content change. Consequently, even with tailored text-to-image alignment feedback fine-tuning, ImageReward achieves a mere 0.004 CLIP score improvement with SD1.5. Therefore, _it is more reasonable to asses the relative improvements_.  Specifically, UniFL obtains 2$\times$ CLIP score improvement over SD1.5 base(0.005 _vs_ 0.01) than the second best method(i.e. SD15-DS), which showcases the superiority of our method. There is also a similar case of other metrics, in which UniFL obtains relatively more notable improvement upon the base model. Moreover, our method also displays an obvious advantage over other methods by the extensive user study.
> 4. **About multiple perceptual aspects optimization with PeFL**:  We show the results of multi-aspect optimization on style and structure through PeFL in Fig.1 of the global rebuttal PDF. As can be seen, incorporating these two distinct objectives does not hurt the effectiveness of each other. Take the prompt ``a baby Swan, graffiti`` as an example, integrating the style optimization via PeFL upon the base model successfully aligns the image with the target style. Further integrating the structure optimization objective retains the correct style and exhibits more complete structural details(e.g. the feet of the Swan) simultaneously.
>
> We hope our rebuttal can address your concerns.

---

> > ### Comment · Reviewer_nwEj · 2024-08-12
> > **Response to rebuttal**
> >
> > Thank you for your rebuttal. After reviewing your rebuttal, it addressed most of my concerns. I tend to increase my rating to Borderline accept.

---

### Official Review · Reviewer_qtQN · 2024-07-15

**Soundness:** 2
**Presentation:** 2
**Contribution:** 2
**Rating:** 4
**Confidence:** 4

**Summary:**

This paper proposes a framework to enhance visual quality, aesthetic appeal, and inference efficiency using various methods, including perceptual feedback learning, decoupled feedback learning, and adversarial feedback learning. Good experimental results are observed.

**Strengths:**

The concept of perceptual feedback learning is promising, as it can leverage various pretrained expert models to improve learning.
The idea of decoupled feedback learning makes sense, as it allows the model to focus on fine-grained details in the images.

**Weaknesses:**

The author claims that previous works primarily focus on individual problems through specialized designs and proposes a unified framework. However, three different methods are designed to solve different problems. Why is it called unified?

In Line 145, it states that image content is incorporated as an additional condition for guidance. However, as shown in Figure 1, no extra inputs are added to the diffusion model.

In the experiment, only an instance segmentation model is used for perceptual feedback learning. Have the authors tried other types of models? It would be interesting to see some analysis on this aspect.

For decoupled feedback learning in Equation 5, why is a hinge loss used instead of a winner-loser loss as in Equation 4? What do the data annotations look like? How to choose the hyper-parameters of the hinge loss?

What are the details of the semantic-based prompt filter and nearest neighbor prompt compression described in Line 189 for active prompt selection?

In Line 202, it states that samples with low inference steps tend to be too noisy to obtain correct rewarding scores in previous methods. How does the proposed adversarial feedback learning improve this and how does it accelerate inference?

**Questions:**

see weakness

**Limitations:**

see weakness

---

> ### Author Rebuttal · Authors · 2024-08-05
>
> Thanks for your kind words about the design of perceptual feedback learning and decoupled feedback learning in our method. We would like to answer the proposed questions in the following:
>
> 1. **Clarification on unified design**:  We claim our method is a unified design as all the modules are seamlessly integrated under the unified feedback learning framework. Specifically, under this framework, we developed different reward mechanisms (namely perceptual reward, decoupled reward, and adversarial reward), which are effectively combined to mitigate the problems of low fidelity, poor appeal, and inefficient inference of existing LDMs simultaneously at the first time via the **_unified rewarding then fine-tuning paradigm_**. In contrast, existing works exhibit high heterogeneity in the way(e.g. changing network structure, or altering the sampling scheduler) to address these defects of LDM, making it challenging for these methods to be well combined to achieve comprehensive improvement.
> 2. **Additional image condition**: We emphasize that _the introduced additional condition of PeFL refers to the image content incorporated into the noise initialization_, instead of the extra control image input like in ControlNet. Specifically, we highlight the perceptual feedback learning(PeFL) with additional image input in Fig.1(left) via the **blue solid line**. As depicted in Fig.1, this image is first injected noise after encoded(the injection process is not illustrated in Fig.1 for clarity, but detailed in Algo.1), then sent to the diffusion model for denoising, the denoised image is then supervised by the perceptual label (instance/semantic seg mask) of the input image. As a comparison, below the input image for PeFL, the decoupled feedback learning starts with pure noise as indicated in the yellow solid line.
> 3. **More instantiation of PeFL**: We have presented two more case studies of perceptual feedback learning with other types of perceptual models, including the semantic segmentation model(i.e. DeepLab-V3) and the style extraction model(i.e. VGG-16) in the Appendix attached to our main paper. The results(Fig.8 and Tab.2 for style, Fig.9 and Fig.10 for layout) demonstrate that PeFL can effectively exploit the prior image knowledge embedded in these expert models to boost the performance of style generation(increasing the style response rate) and semantic layout optimization(more preferred layout).
> 4. **Loss formulation in Eq.5**: We choose the hinge loss instead of the winner-loser loss in Eq.5, as _it describes the reward-tuning procedure, wherein there are no winner and loser samples for pair-wise loss calculation_. Specifically, in this process, a sample is generated from the pure noise given a prompt and then scored via the already well-trained reward models. Eq.5 then encourages the generated image to gain a higher reward score, and the hinge coefficient helps to avoid the unbounded increasing reward, preventing the LDM overfit to the reward model. The winner-loser loss is only used in the reward model training stage, where the reward model is trained to increase the score of the preferred sample while lowering the unpreferred one, aligning with the human preference.
> 5. **Preference data annotations**: We provide some examples of the collected annotated preference data in Sec.B.1(Fig.11) in the Appendix of our main paper.
> 6. **More details of active prompt selection**: We provide more details of the semantic-based prompt filter and nearest neighbor prompt compression during active prompt selection in Sec. B.2 in the Appendix of the main paper. In short, the semantic-based prompt filter selects the prompts owning rich semantic meaning by the ruled-based filtering, while the nearest neighbor prompt compression is designed to suppress the redundant prompts by checking the semantic similarity in the embedding space to ensure diversity.
> 7.  **Selection of hinge coefficient $\alpha_d$**: Please refer to the Author Rebuttal part and Fig.4  of the global rebuttal PDF for more details.
> 8. **Analysis on Adavasarial Feedback Learning**: We provide a detailed analysis of the effect of adversarial feedback learning on acceleration in our ablation study section (L316). In summary, incorporating adversarial feedback plays two roles: (1) With the diffusion model and the adversarial reward model updated adversarially, the diffusion model is not prone to overfitting. The synchronously updated discriminator forces the diffusion model to evolve continuously, enjoying the reward guidance for a longer duration; (2) The denoised images under the low inference steps are forced to be clearer via the strong adversarial objective and thus can be correctly rewarded by the aesthetic reward model. Given these two benefits, the generation quality of low-step images steadily improved via the reward guidance, achieving inference acceleration.
>
> We hope our rebuttal can address your concerns.

---

### Author Rebuttal · Authors · 2024-08-06

We sincerely thank all the reviewers for their constructive comments. We are delighted that the core contributions of our proposed method are regarded as promising and effective (R-qtQN, R-dHXQ, R-af1S), novel (R-dHXQ), the paper is well-motivated (R-dHXQ) and well-written (R-nwEj, R-dHXQ, R-af1S). All the comments will be addressed in the revised paper. We would like to answer some common questions in the following:

**R-qtQN, R-af1S: Selection of hinge coefficient $\alpha_d$**: We customized the hinge coefficient for each aesthetic reward model based on their reward distributions on the validation set. As illustrated in Fig.4(left) of the global rebuttal PDF,  there are clear margins in the reward scores between preferred and unpreferred samples. Moreover, such margin varies across these dimensions, which emphasizes the importance of the decoupled design. Empirically, we set the reward hinge coefficient to the average reward scores of the preferred samples to encourage the diffusion model to generate the sample with higher reward scores. Taking the reward model on color as an example, we ablate such coefficient selection. As depicted in Fig. 4(right) of the global rebuttal PDF, setting $\alpha_{color}$ too small resulted in marginal improvement due to limited guidance, while excessively large hinge coefficients led to over-saturation in images. Conversely, selecting a coefficient around the average reward score of the preferred samples yielded optimal results. A similar trend was observed in the layout and lighting aesthetic dimensions, except for the detail dimension. Notably, a slightly lower $\alpha_{detail}$ sufficed to achieve satisfactory results for the detail reward model, whereas a higher coefficient tended to introduce more background noise. This phenomenon is likely attributed to the substantial reward score margin between preferred and unpreferred samples for detail dimension, where a high coefficient could lead to overwhelming guidance toward the target reward dimension.

We hope our rebuttal can address the concerns about these questions.

---

### Comment · Area_Chair_H5NP · 2024-08-08
**Kindly reminder to respond to author responses**

Dear Reviewers,

Thank you very much again for performing this extremely valuable service to the NeurIPS authors and organizers.

As the authors have provided detailed responses, it would be great if you could check them and see if your concerns have been addressed. Your prompt feedback would provide an opportunity for the authors to offer additional clarifications if needed.

Cheers,

AC

---

> ### Comment · Area_Chair_H5NP · 2024-08-13
>
> Dear Reviewers,
>
> As the discussion phase is ending in one day, it would be helpful if you could respond to the authors' rebuttal if you haven’t already. This would allow the authors to address any remaining concerns.
>
> Thanks,
>
> AC

---

### Decision · Program_Chairs · 2024-09-25

**Decision:**

Accept (poster)

**Comment:**

The paper initially received mixed reviews. The reviewers appreciated the idea of introducing a unified feedback learning framework capable of integrating various priors from perceptual models to enhance different diffusion models, as well as the extensive experiments conducted. However, they expressed concerns about the limited novelty compared to previous work like ReFL, the unconvincing performance improvements and qualitative results, and the lack of explanations for some key design choices, implementation details, ablation studies, and user studies.


The rebuttal partially addressed the reviewers’ concerns by providing additional explanations, experimental results, and discussions. After the rebuttal, Reviewers dHXQ, nwEj, and af1S rated the paper as borderline accept or weak accept, while Reviewer qtQN maintained a borderline reject score. During the subsequent discussion, Reviewers dHXQ, nwEj, and af1S acknowledged that the rebuttal had addressed their concerns. However, Reviewer qtQN did not participate in the discussion and did not update their final recommendation.


The AC has read the paper, reviews, and rebuttal, and discussed with the reviewers at length. The AC found the arguments of Reviewers dHXQ, nwEj, and af1S to be the most convincing, noting that the concerns raised by Reviewer qtQN were either addressed in the original submission or adequately explained in the rebuttal. As a result, the AC weighed the paper's strengths more heavily than its current weaknesses.


The authors are encouraged to improve the camera-ready version by following the reviewers' recommendations, particularly by including the new experimental results provided in the rebuttal, offering a more in-depth discussion of the differences from prior work, and adding further explanation and analysis. Such revisions would greatly enhance the work and significantly increase its impact. This decision was discussed with and approved by the AC and SAC.